# Synthesis of Bithiophene-Based D-A_1_-D-A_2_ Terpolymers with Different A_2_ Moieties for Polymer Solar Cells via Direct Arylation

**DOI:** 10.3390/polym11010055

**Published:** 2019-01-02

**Authors:** Jinfeng Huang, Zhenkun Lin, Wenhuai Feng, Wen Wang

**Affiliations:** Fujian Key Laboratory of Polymer Materials, College of Chemistry and Materials Science, Fujian Normal University, Fuzhou 350007, China; hjf@fjnu.edu.cn (J.H.); 18005978449@163.com (Z.L.); whfeng25@126.com (W.F.)

**Keywords:** D-A_1_-D-A_2_ terpolymers, direct arylation, polymer solar cells, pyrrolo[3,4-*c*]pyrrole-1,4-dione, bithiophene

## Abstract

A series of bithiophene (2T)-based D-A_1_-D-A_2_ terpolymers with different A_2_ moieties were prepared via direct arylation reaction. In these terpolymers, pyrrolo[3,4-*c*]pyrrole-1,4-dione (DPP) was selected as the first electron-accepting (A_1_) moiety, 2,1,3-benzothiadiazole (BT) or fluorinated benzothiadiazole (FBT) or octyl-thieno[3,4-*c*]pyrrole-4,6-dione (TPD) or 2,1,3-benzoselendiazole (SeT) was selected as the second electron-accepting (A_2_) moiety, while bithiophene with hexyl side chain was used as the electron-donating moiety. The UV-vis absorption, electrochemical properties, blend film morphology, and photovoltaic properties were studied to explore the effects of the A_2_ moiety. It is shown that these terpolymer films exhibit broad absorption (350–1000 nm), full width at half-maximum of more than 265 nm and ordered molecular packing. Varying the A_2_ moiety could affect the energy levels and blend film morphology leading to different polymer solar cell (PSC) performances of these (2T)-based D-A_1_-D-A_2_ terpolymers. As a result, the highest *J_sc_* of 10.70 mA/cm^2^ is achieved for Polymer 1 (P1) with BT as A_2_ moiety, while the higher highest occupied molecular orbital (HOMO) level limits the open circuit voltage (*V_oc_*) and leads to a power conversion efficiency (PCE) of 3.46%.

## 1. Introduction

Over the past decade, continuous efforts have been devoted to polymer solar cells (PSCs) [1,2]. Development and impressive power conversion efficiency (PCE) values in the range of 10–14% were achieved [3,4,5]. These achievements were mainly attributed to the advance in electron-rich unit (D)-electron-accepting unit (A) alternating low band gap polymers. In addition, the development of the single-material organic solar cell (SMOC) with high efficiency has been reported by the use of electro-spinning to obtain ultrathin fibers with aligned D-A polythiophene chains [6]. However, newly synthesized D-A copolymers typically have narrow absorption bands with a full width at half-maximum of 200 nm, which is not very broad [7,8]. Recently, the use of regio-regular D-A_1_-D-A_2_ terpolymers as polymer donors, which contain three components with one electron-donating unit and two different electron-accepting units on their conjugated polymer backbones, provides an effective approach for tuning optical and electrochemical properties, thereby enhancing the photovoltaic performance of PSCs. Therefore, developing high-performance regio-regular D-A_1_-D-A_2_ terpolymer donors for application in solar cells is in demand [9,10,11,12,13]. Furthermore, these polymers could be potential materials for field-effect transistors (OFETs) and light-emitting diodes (PLED) [14]. Most importantly, conjugated polymers became extensively studied for biomedical application, especially for photothermal therapy (PPT) and its combination with other therapies [15].

How to design and synthesize conjugated polymers by simpler, faster, and cheaper means is an important research area. Pd-catalyzed C−H direct arylation (DA) reaction is an emerging, advantageous tool for the synthesis of conjugated polymers with a large variety of monomers. This methodology has some obvious advantages, such as fewer synthetic steps, less time, less harmful substance [16,17,18,19,20]. In our previous work, novel terthiophene (3T)-based D-A_1_-D-A_2_ terpolymers with terthiophene as electron-donating units by DA, have been obtained and applied for PSCs [21]. However, the molecular weight of these 3T-based polymers was not high due to the steric hindrance of the monomer. Many works have revealed that the molecular weight of the polymer has positive influence on the PSC performance [22,23]. Compared with 3T-based terpolymers, bithiophene (2T)-based terpolymers with higher M_n_ can be easily obtained by DA polymerization. Our previous work demonstrated that the efficient bithiophene (2T)-based terpolymers can be simply synthesized [24]. As well known, electron-accepting moiety can obviously influence the packing structure and properties of polymers. However, few reports have studied effects of secondary acceptor moiety (A_2_) on the properties of 2T-based D-A_1_-D-A_2_ terpolymers obtained by DA polymerization.

Herein, we designed synthesis of bithiophene (2T)-based terpolymers with different A_2_ moieties via DA reaction. Here, D-A_1_-D and D-A_2_-D intermediates were synthesized as the building blocks. In these building blocks, pyrrolo[3,4-*c*]pyrrole-1,4-dione (DPP) was selected as the first electron-accepting moiety (A_1_), 2,1,3-benzothiadiazole (BT), or fluorinated benzothiadiazole (FBT), or alkylthieno[3,4-*c*]pyrrole-4,6-dione (TPD), or 2,1,3-benzoselendiazole (SeT) was selected as the secondary electron-accepting moiety (A_2_) [25,26,27,28]. Bithiophene moiety with hexyl side chain was selected as the donor unit. Then, four 2T-based D-A_1_-D-A_2_ terpolymers were synthesized with D-A_1_-D and D-A_2_-D monomers via DA in fewer synthetic steps compared to the reported D-A_1_-D-A_2_ terpolymers via Stille or Suzuki coupling methods [29,30]. The UV-vis absorption, electrochemical properties, film morphology, and photovoltaic properties were studied to explore the effects of the secondary electron-accepting moiety.

## 2. Experimental Section

### 2.1. Materials

Monomer 2,5-bis(2-octyldodecyl)-3,6-di(thiophen-2-yl)pyrrolo[3,4-*c*]pyrrole-1,4(*2H*, 5*H*)-dione (D-A_1_-D, purity: >98%) was purchased from Derthon, Three monomers (D-A_2_-D), 4,7-bis(5-bromo-4-hexylthiophen-2-yl)benzo[*c*][1,2,5]thiadiazole(A_2_: BT), 4,7-bis(5-bromo-4-hexylthiophen-2-yl)-5,6-difluorobenzo[*c*][1,2,5]thiadiazole (A_2_: FBT), 1,3-bis(5-bromo-4-hexylthiophen-2-yl)-5-octyl-4*H*-thieno[3,4-*c*]pyrrole-4,6(5*H*)-dione (A_2_: TPD) were synthesized by direct arylation according to our previous literature [21]. The synthetic detail of 4,7-bis-(5-bromo-4-hexylthiophen-2-yl)-benzo[1,2,5]selenadiazole (A_2_: SeT) was provided in Appendix A. PEDOT:PSS (Baytron PVP A1 4083), palladium acetate, tricyclohexylphosphoniumtetrafluoroborate were purchased from J&K Scientific Ltd. All solvents (*N*,*N*-dimethylacetamide (DMAc), xylene, chloroform, methanol, acetone and hexane) were purchased from Sinopharm and must be distilled in order to exclude water and oxygen in the reaction.

### 2.2. Synthesis of Polymers (P1, P2, P3, P4)

Four bithiophene (2T)-based terpolymers with different A_2_ moieties via DA reaction were synthesized. First, monomer (D-A_1_-D) 2,5-bis(2-octyldodecyl)-3,6-di(thiophen-2-yl)pyrrolo[3,4-*c*]pyrrole-1,4(2H, 5H)-dione (0.1 mmol, 86.1 mg) and either monomer (D-A_2_-D) (4,7-bis(5-bromo-4-hexylthiophen-2-yl)benzo[*c*][1,2,5]thiadiazole or 4,7-bis(5-bromo-4-hexylthiophen-2-yl)-5,6-difluorobenzo[*c*][1,2,5]thiadiazole or 1,3-bis(5-bromo-4-hexylthiophen-2-yl)-5-octyl-4H-thieno[3,4-*c*]pyrrole-4,6(5H)-dione or 4,7-bis-(5-bromo-4-hexylthiophen-2-yl)-benzo[1,2,5]selenadiazole)) (0.1 mmol), as well as potassium carbonate (0.25 mmol), pivalic acid (0.1 mmol), palladium acetate (Pd(OAc)_2_) (0.005 mmol) and tricyclohexylphosphoniumtetrafluoroborate (0.01 mmol) were mixed in 1 mL DMAc /p-xylene (V/V = 1:1) under nitrogen. Then, the mixture was heated to reflux for 48 h. The mixture was cooled down and precipitated from methanol. The product was sequentially purified via Soxhlet extraction using methanol, acetone, hexane, and chloroform. The chloroform fraction was concentrated and re-precipitated from the methanol. The final dark-black solid was obtained by drying under vacuum overnight.

Polymer P1: yield: 75.9%. Gel permeation chromatography (GPC) (tetrahydrofuran (THF), polystyrene standard): number-average molecular weight (M_n_) = 32.4 kg/mol, weight-average molecular weight (M_w_) = 64.7 kg/mol, polydispersity index (PDI) = 1.99. ^1^H NMR (400 MHz, CDCl_3_) δ: 9.11–8.72 (br, 2H), 8.10–7.56 (br, 4H), 6.98 (br, 2H), 4.10–3.56 (br, 4H), 2.80–0.56 (br, 100H). Polymer P2: yield: 43.8%. GPC (THF, polystyrene standard): M_n_ = 26.7 kg/mol, M_w_ = 42.2 kg/mol, PDI = 1.58. ^1^H NMR (400 MHz, CDCl_3_) δ: 9.05–8.70 (br, 2H), 8.10–7.65 (br, 2H), 6.98 (br, 2H), 4,12–3.78 (br, 4H), 2.75–0.56 (br, 100H). Polymer P3: yield: 49.2%. GPC (THF, polystyrene standard): M_n_ = 40.5 kg/mol, M_w_ = 62.7 kg/mol, PDI = 1.55. ^1^H NMR (400 MHz, CDCl_3_) δ: 8.98–8.50 (br, 2H), 8.05–7.50 (br, 2H), 7.03–6.50 (br, 2H), 4.01–3.42 (br, 6H), 2.82–0.82 (br, 119H). Polymer P4: yield: 34.4%. GPC (THF, polystyrene standard): M_n_ = 42.2 kg/mol, M_w_ = 62.2 kg/mol, PDI = 1.47. ^1^H NMR (400 MHz, CDCl_3_) δ: 9.10–8.80 (br, 2H), 8.01–7.51 (br, 4H), 7.10–6.80 (br, 2H), 4,04–3.81 (br, 4H), 2.86–0.83 (br, 100H).

### 2.3. Measurement

The NMR spectra were performed on a BRUKER AVIII-400 NMR spectrometer. The molecular weights were measured using water 1515 gel permeation chromatography (GPC) analysis using tetrahydrofuran column and tetrahydrofuran as eluent. The GPC was calibrated against polystyrene standard. Thermogravimetric analysis (TGA) was used to measure the thermal property on analysis system (Mettler-Toledo 851e/822e) under nitrogen, at a heating rate of 10 °C/min. Uv-vis absorption spectra were performed on a SHIMADZU UV-2600 spectrometer. Cyclic voltammetric (CV) measurements were carried out on a Zahner IM6e Electrochemical Workstation in a tetrabutylammonium hexafluorophosphate (Bu4NPF6) (0.1 M) acetonitrile solution. Polymer films were drop cast onto the working electrode, a platinum wire was the counter electrode, and an Ag/Ag^+^ electrode was the reference electrode. The scan rate was set to 20 mV s^−1^. Atomic force microscopy (AFM) images of the thin films were carried out on a NanoMan VS microscope using a BRUKER RTESP-300 probe (operation mode: tapping; material: 0.01–0.025 ohm-cm Antimony (n) doped Si; spring constant: 40 N/m; scanning rate: 0.6–1 μm/s).

### 2.4. Fabrication of Devices and Characterization

Photovoltaic devices were fabricated with a conventional sandwich structure of ITO/PEDOT:PSS/polymers:PC_71_BM/PFN/Al. in a N_2_-filled glove box. The ITO-coated glass and PEDOT:PSS (30 nm) were treated as described in the literature. Then the blends comprising of polymer:PC_71_BM were dissolved in chloroform with 3% 1,8-diiodooctane (DIO) and the solution concentration was 15 mg/mL. The blended solutions were spin-coated on the ITO/PEDOT:PSS substrate to generate the active layer. Afterwards, poly[(9,9-dioctyl-2,7-fluorene)-alt-(9,9-bis(3-(*N*,*N*-dimethylamino)propyl)-2,7-fluorene)] (PFN) solution (0.4 mg/mL) was spin-coated at 3500 rpm on the surface of active layer. Finally, the Al electrode (100 nm) was successively deposited under a high vacuum on top of the PFN layer. The active areas of the cells were 4 mm^2^. A Keithley 2400 Source Meter under 100 mW cm^2^ simulated AM 1.5 G irradiation was used to measure current-voltage (J-V) characteristics. Enlitech QE-R3011 at room temperature was used to measure the external quantum efficiency (EQE) values of the devices.

## 3. Results and Discussion

### 3.1. Synthesis and Thermal Property

We synthesized four intermediate monomers (D-A_2_-D); {4,7-bis(5-bromo-4-hexylthiophen-2-yl)benzo[*c*][1,2,5]thiadiazole (A_2_: BT), 4,7-bis(5-bromo-4-hexylthiophen-2-yl)-5,6-difluorobenzo[*c*][1,2,5]thiadiazole (A_2_: FBT), 4,7-bis(5-bromo-4-hexylthiophen-2-yl)-5-octyl-4*H*-thieno[3,4-*c*]pyrrole-4,6(5*H*)-dione (A_2_: TPD) according to our previous literature [21]. The synthetic detail of 4,7-bis-(5-bromo-4-hexylthiophen-2-yl)-benzo[1,2,5]selenadiazole (A_2_: SeT) was provided in Appendix A. The bithiophene (2T)-based D-A_1_-D-A_2_ terpolymers were prepared via DA polymerization, in which DPP (A_1_)-based monomer (D-A_1_-D) (2,5-bis(2-octyldodecyl)-3,6-di(thiophen-2-yl)pyrrolo[3,4-*c*]pyrrole-1,4(2H, 5H)-dione) was a C-H monomer and the second acceptor (A_2_)-based monomer (D-A_2_-D) was a C-Br monomer. Scheme 1 shows the synthetic route of the monomers and polymers. In this work, the hexyl side chain was incorporated into thiophene moiety to construct a C-Br monomer to promote the DA polymerization and solubility [31]. Under nitrogen atmosphere, by way of the DA condition; K_2_CO_3_ (0.25 mmol), pivalic acid (0.1 mmol), catalyst (Pd(OAc)_2_) (0.005 mmol) and ligand (tricyclohexylphosphoniumtetrafluoroborate) (0.01 mmol), the ultimate target 2T-based D-A_1_-D-A_2_ terpolymers with high molecular weight (M_n_ values range from 27 to 42 kg/mol) were successfully obtained. It demonstrated that 2T-based D-A_1_-D-A_2_ terpolymers in this work displayed improved M_n_ values in comparison with previously reported 3T-based terpolymers (M_n_ values ranged from 12 to 24 kg/mol) [21]. Compared to the C-H monomer (D-A_1_-D) in this work, monomer 2T-DPP-2T that used as C–H monomers to construct 3T-based terpolymers in our previous work, has more thiophene moieties, resulting in negative steric influence during DA polymerization. Therefore, bithiophene (2T)-based terpolymers with high M_n_ value can be easily obtained. All polymers were subjected to sequential Soxhlet extraction with methanol, acetone, hexane, and chloroform. The chloroform fractions were concentrated under reduced pressure and precipitated in methanol to obtain the resulting polymers.

All these polymers show good thermal stability, as revealed by thermogravimetric analysis (TGA) with decomposition temperature over 350 °C. As shown in Figure 1, the onset decomposition temperature (T_d_) (5% weight loss) was found to be at 387 °C (P1), 368 °C (P2), 385 °C (P3) and 386 °C (P4), respectively. Good thermal stability is helpful for stabilized morphology of the active layer at elevated temperatures.

### 3.2. Optical and Electrochemical Properties

The optical properties of polymers were characterized by the UV-vis spectra of dilute (1 × 10^−6^ mol/L) chloroform solutions and of thin films. The concentration for monomer unit D-A_1_-D-A_2_ was 2.4 (P1), 2.0 (P2), 2.8 (P3), 3.1 (P4) × 10^−5^ mol/L in solution. The absorption maxima wavelength (λ_max_), optical band gap deduced from the solution and film absorption is summarized in Table 1. As shown in Figure 2, all these 2T-based D-A_1_-D-A_2_ terpolymers showed broad absorption (350–1000 nm) and full width at half-maximum of more than 265 nm in solutions and thin films. This is the typical advantage of these D-A_1_-D-A_2_ terpolymers. The broader absorption was beneficial to capture more sunlight to generate photocurrent. In solution, all polymers showed a strong and broad absorbance band at the range of 350–1000 nm. This was attributed to the intra-molecular charge transfer (ICT) from the donor moiety to the acceptor moiety in the polymer backbone. The weak absorbance band, at the range 350–450 nm, that was observed in P3, originated from the π–π* transition of polymer backbones [32]. When replacing BT with FBT or TPD or SeT as the A_2_ moiety, a blue-shifted absorbance band was observed. From solution to film, polymer P1 and P2 afforded a slightly blue-shifted and narrowing absorbance band due to the formation of molecular packing and solvent effect in the polymer (P1 or P2) solution. The other two terpolymers exhibited broadened and red-shifted absorbance bands due to the stronger intermolecular interactions and the extension of the effective conjugated length in condensed solid states. In addition, absorbance bands (350–450 nm) with low intensity were observed in all polymer films, which demonstrated that the π–π* transition of polymer backbones was strengthen in the solid state. The shoulder peaks at the longer wavelength were obviously observed in polymer films. These demonstrated that the π–π stacking interaction of ordered molecular packing existed in the solid state. The fluorinated polymer P2 displayed a clear blue-shift compared to the non-fluorinated polymer P1, while no similar phenomena were observed in our reported 3T-polymer [21]. When replacing BT with TPD or SeT as the A_2_ moiety, polymer P3 and P4 films displayed a similar spectrum compared to the P1 film. From the onset of film absorption, the optical bandgaps of P1–P4 were estimated to be 1.23, 1.26, 1.22, 1.22 eV, respectively, which demonstrated that they were the narrow bandgap polymers.

Cyclic voltammetry (CV) measurement of the polymers was performed to explore the effects of A_2_ moieties on their electrochemical properties. The highest occupied molecular orbital (HOMO) and the lowest unoccupied molecular orbital (LUMO) levels of the polymers were calculated by the empirical relationship [33]. E_HOMO_ = −e(E_oxd_−E_Fc/Fc_^+^ + 4.80)V, E_LUMO_ = −e(E_red_ − E_Fc/Fc_^+^ + 4.80)V, where the formal potential of Fc/Fc^+^ was measured as 0.38 V against Ag/Ag^+^ [34].

The CV curves and energy levels of the polymers are provided in Figure 3. In order to reduce the errors caused by the broad and irreversible oxidation peaks observed in the polymer CV curves, the onset oxidation (E_oxd_) and reduction potentials (E_red_) were obtained by using tangent method. The respective HOMO and LUMO energy levels are summarized in Table 1. The HOMO and LUMO energy levels of fluorinated polymer P2 were lower than that of non-fluorinated polymer P1 due to the electron-withdrawing nature of the fluorine atoms. As the A_2_ moiety changed from BT to SeT, the energy levels were similar. As the A_2_ moiety changed from BT to TBD, the energy levels were decreased. The deeper HOMO levels may obtain higher *V_oc_*. The results showed that varying the A2 moiety could influence the energy levels of the obtained D-A_1_-D-A_2_ polymers. However, the LUMO offset between fluorinated terpolymer P2 (−3.72 eV) and PC_71_BM (−4.0 eV) was only 0.28 eV, which might not provide a sufficient LUMO offset to ensure highly efficient exciton dissociations for polymer: PC_71_BM cells.

### 3.3. Photovoltaic Properties

In order to explore the effects of the A_2_ moiety on the photovoltaic properties, four polymers were paired with PC_71_BM to fabricate the devices with the conventional architecture of ITO/PEDOT:PSS/active layer/PFN/Al. Figure 4A,B shows the current density-voltage (J-V) curves and the external quantum efficiency (EQE) of these devices. The summary of device performance parameters for the device based polymer: PC_71_BM are listed in Table 2.

As shown in Figure 4A and Table 2, the PCE of the four 2T-based terpolymers strongly depended on the secondary electron-accepting (A_2_) moiety. For P1, with BT as the A_2_ moiety, the highest J_sc_ values were obtained among the devices based four polymers. Compared to our reported polymer with the dodecyl side chain under the similar molecular weight [24], P1 with the hexyl side chain displayed enhanced *J_sc_* (9.34→10.70 mA/cm^2^). This is owing to the more ordered packing structure of the polymer with its shorter side chain. The highest PCE of P1 was found to be 3.46%, with a *V_oc_* of 0.59 V, a *J_sc_* of 10.70 mA/cm^2^ and a FF of 54.8%. Compared to P1, the PCE of P2 was obviously reduced, the PCE of 0.5%, with a *V_oc_* of 0.61 V, a *J_sc_* of 2.32 mA/cm^2^ and a FF of 35.4%. Therefore, changing the A_2_ moiety from BT to FBT significantly reduced the *J_sc_* and *FF*. The reduction in *J_sc_* might be related to the E_LUMO_^DONOR^–E_LUMO_^PC^_71_^BM^ energy level offset [35]. For P2, the E_LUMO_ offset with PC_71_BM was 0.28 eV, which was below the 0.3 eV. In addition, the bad morphology of active layer is the likely factor in the lower observed *J_sc_* and FF in P2. For P3, as the A_2_ moiety replacing BT with TPD, the *V_oc_* and FF were obviously enhanced, to 0.72 V and 65.3%, but the *J_sc_* was reduced to 5.93 mA/cm^2^, resulting in a PCE of 2.79%. Compared to P1, P4 with SeT as the A_2_ moiety, had a decreased *J_sc_* of 7.08 mA/cm^2^ and FF of 47.2%. The good charge transport ability and morphology of active layer could contribute to the improved *J_sc_* of the device based P1.

External quantum efficiencies (EQEs) were measured for different PSC devices under optimized device conditions (Figure 4B). As shown in Figure 4B, the EQE spectra of the four polymers showed a clear photo-response over the spectral region between 300–900 nm, consistent with the absorption of PC_71_BM and the polymer. Device based P1 showed the highest photo-response with a maximal EQE in a broad range of 350–900 nm, that benefited from the intensive absorption of polymer P1 in this region. This result is in accordance with the highest J_sc_ of the corresponding device based P1. In contrast, the device based fluorinated-polymer P2 showed negligible contribution in the absorption of polymer PC_71_BM, leading to the lowest *J_sc_*. However, the peak positions of EQE spectra were different from the absorption spectra of polymers because the photocurrents were contributed by the PC_71_BM (300–500 nm) and polymer (500–900 nm) absorption.

### 3.4. Active Layer Morphology

The morphological properties of the four types of donor:PC_71_BM blend films were estimated by atomic force microscopy (AFM) to investigate the correlation between the morphology and the PCE of these four polymers. Figure 5 and Appendix A show the AFM topographic and 3D images of the blend films. As shown in Appendix A, the surface roughness (RMS) of the blend film was 2.62 nm (P1:PC_71_BM), 11.3 nm (P2:PC_71_BM), 4.71 nm (P3:PC_71_BM), 3.86 nm (P4:PC_71_BM), respectively. As shown in Figure 5, the P1:PC_71_BM film exhibited relatively uniform and smooth morphology with less aggregation, which is beneficial for charge carrier transport. In addition, the proper and continuous phase separation was observed in the P1:PC_71_BM film, which could assure efficient exciton dissociation. This was in accordance with the highest *J_sc_* obtained in the device based P1. In contrast, the P2:PC_71_BM film exhibited distinct globular aggregate and rough surface, which might originate from the poor miscibility of the polymer P2 with PC_71_BM. The large domain phase separation in P2 blend films limited the photocurrent due to inefficient charge separation of the active layer. Furthermore, an insufficient energy offset between P2 and PC_71_BM could also limit the photocurrent, resulting in the lowest *J_sc_* for the device based fluorinated polymer P2. In addition, compared to P1, P3 (P4), PC_71_BM blend films exhibited coarser and rougher morphology, which might be the reason for the lower *J_sc_* for the device based P3 (P4).

## 4. Conclusions

In summary, a synthetic procedure for a series of bithiophene (2T)-based D-A_1_-D-A_2_ terpolymers with different A_2_ moieties using DA was provided. This synthesis of 2T-based D-A_1_-D-A_2_ terpolymers is non-toxic and simple, compared with existing routes [29,30]. It was shown that the A_2_ moiety could introduce difference in the energy levels, packing structure, blend film morphology and PSC performance of the polymers. Finally, the highest *J_sc_* of 10.70 mA/cm^2^, was achieved for P1 with BT as the A_2_ moiety, while the higher HOMO level limits the open circuit voltage (*V_oc_*) and leads to a power conversion efficiency (PCE) of 3.46%. The results demonstrate that varying the electron-accepting moiety can be an effective strategy for the construction of D-A_1_-D-A_2_ terpolymers.

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
