# Peer review of "Synthesis of Bithiophene-Based D-A1-D-A2 Terpolymers with Different A2 Moieties for Polymer Solar Cells via Direct Arylation"

_polymers, 2019, doi:10.3390/polym11010055_

Reviewer 1 Report

In this manuscript, W. Wen et al. report bithiophene-based D-A1-D-A2 type terpolymers for polymer solar cells. Four different acceptor moieties as A2 were investigated and compared. Similar studies were previously reported by the authors group for terthiophene-based polymers. One of the acceptors (BT) in the same polymer structure was also already reported. This work is an extension of these previous works and give some variations. Therefore, the novelty of the present manuscript is low but the work could be publishable with three new polymers as applied to polymer solar cells. However, I have some concerns to be addressed before the acceptance of the manuscript.

1) The most basic and important point is the reliability/reproducibility of the solar cell performance data. How many devices have been fabricated for each polymer and what are the variations in the performance parameters given in Table 2.

2) The authors states that a motivation of the present work comes from a problem associated with the terthiophene-based polymer that the molecular weights were not high. However, there is no mention about how this problem was resolved by preparing the bithiophene-based polymers in the present work. This should be described by comparing the molecular weights for these polymers.

3) The D-A2-D fragment with A2 being SeT was not reported in ref. 18. If this is a novel compound full synthetic details and characterization data should be given.

4) Line 161: "From solution to film, all the four terpolymers exhibited broadened and red-shifted peaks" - This is not correct! It is obvious from Fig. 2 that some polymers (P1 and P2) show blue-shifted absorption on going from the solution to the film.

5) Paragraph starting from line 186: The HOMO and LUMO levels must have relatively large uncertainties because of the broad, irreversible oxidation peaks in CV (Fig. 3). These large uncertainties must be taken into account in discussing the HOMO/LUMO levels.

6) I would like have a comment from the authors about why the absorption spectra (Fig. 2) and IPCE spectra (Fig. 4) are so different, particularly in terms of the peak positions.

Below are minor points.

7') The full spelling should be given for PSC at the first appearance. Check for every abbreviation/acronym in the manuscript.

8) Line 59 "fewer synthetic steps": Fewer synthetic steps that what?

9) Scheme 1: Connectivity between D and A2 in D-A2-D should be explicitly given: e.g., which atoms in BT is bonded to the thiophene moieties in D-A2-D?

10) Line 151: "-6" should be a superscript. By the way, is this concentration for monomer units (D-A1-D-A2-D)? - which should be defined.

11) I do not understand what it means by "relatively highest" which appears a few times in the text.

12) Line 252: What is "the existing routes". At least give a reference.

Author Response

1)      The most basic and important point is the reliability/reproducibility of the solar cell performance data. How many devices have been fabricated for each polymer and what are the variations in the performance parameters given in Table 2.

Thanks for the reviewer’s kind suggestion. According to the suggestion, we have supplied the average values from 10 devices fabricated for each polymer in Table 2 and marked in red.

2)      The authors states that a motivation of the present work comes from a problem associated with the terthiophene-based polymer that the molecular weights were not high. However, there is no mention about how this problem was resolved by preparing the bithiophene-based polymers in the present work. This should be described by comparing the molecular weights for these polymers.

 Thanks for the reviewer’s kind suggestion. According to the suggestion, we have provided the comment by comparing the molecular weights in line 149-153 and marked in red.

3)      The D-A2-D fragment with A2 being SeT was not reported in ref. 18. If this is a novel compound full synthetic details and characterization data should be given.

Thanks for the reviewer’s kind suggestion. According to the suggestion, we have provided the synthetic details and characterization data of the monomer in supporting information.

4)      Line 161: "From solution to film, all the four terpolymers exhibited broadened and red-shifted peaks" - This is not correct! It is obvious from Fig. 2 that some polymers (P1 and P2) show blue-shifted absorption on going from the solution to the film.

Thanks for the reviewer’s kind suggestion. According to the suggestion, we have modified Fig.2A and B to align X-axis. We checked the data and spectra carefully, then found P1 and P2 showed slightly blue-shifted absorption on going from the solution to the film. The corresponding description are provided in line 176-178 and marked in red.

5)      Paragraph starting from line 186: The HOMO and LUMO levels must have relatively large uncertainties because of the broad, irreversible oxidation peaks in CV (Fig. 3). These large uncertainties must be taken into account in discussing the HOMO/LUMO levels.

Thanks for the reviewer’s kind suggestion. According to the suggestion, we have discussed in line 206-208 and marked in red.

6)      I would like have a comment from the authors about why the absorption spectra (Fig. 2) and IPCE spectra (Fig. 4) are so different, particularly in terms of the peak positions.

Thanks for the reviewer’s kind suggestion. According to the suggestion, we have added the corresponding comments in line 253-255 and marked in red.

Below are minor points.

  7) The full spelling should be given for PSC at the first appearance. Check for every abbreviation/acronym in the manuscript.

    Thanks for the kind suggestion. We have checked the manuscript and given the full spelling for the abbreviation/acronym at the first appearance.

8)      Line 59 "fewer synthetic steps": Fewer synthetic steps that what?

Thanks for the reviewer’s kind suggestion. According to the suggestion, we have addedcompared to the reported D-A1-D-A2 terpolymers via Stille or Suzuki coupling methods” in line 64-65.

9)      Scheme 1: Connectivity between D and A2 in D-A2-D should be explicitly given: e.g., which atoms in BT is bonded to the thiophene moieties in D-A2-D?

Thanks for the reviewer’s kind suggestion. We have revised the scheme1 according to the suggestion.

10)      Line 151: "-6" should be a superscript. By the way, is this concentration for monomer units (D-A1-D-A2-D)? - which should be defined.

Thanks for the reviewer’s kind suggestion. According to the suggestion, we have revised and defined the monomer unit in line 164-166 and marked in red.

  11) I do not understand what it means by "relatively highest" which appears a few times in the text.

     Thanks for the reviewer’s kind suggestion. We have deleted the “relatively” according to the suggestion.

   12) Line 252: What is "the existing routes". At least give a reference.

     Thanks for the reviewer’s kind suggestion. According to the suggestion, we have added the cited reference [reference 29, 30].

Reviewer 2 Report

Summary 

The manuscript entitled “Synthesis of bithiophene-based D-A1-D-A2 terpolymers with different A2 moieties for polymer solar cells via direct arylation” by Jinfeng et al. shows the synthesis of bithiophene (2T)-based terpolymers with different A2 moieties via DA reaction. The synthesized polymers have been finally used to fabricate organic solar cells

The polymers have been fully chemically, optically, and thermally characterized. Moreover, the Authors demonstrated the applicability of these nanostructured materials through photovoltaic studies performed on BHJ cells fabricated blending the D-A polymers with PC71BM.

General comments 

In general, the work is accurate and clearly presented, and the results are of interest to readers of Polymers. The material characterization is detailed and the photovoltaic tests are useful to prove the potential impact of the developed structures.   
The article style is correct but it should be reviewed in a few points. Thus, I believe that the text needs some technical adjustments to be published. Therefore, I recommend that this manuscript can be published in Polymers after major revision. 

Specific comments

The article grammar, punctuation and style are adequate and the manuscript do not need to be proofread. Anyway, there is a problem in the Introduction section. There are a few vital notions that should be given to readers. Considering that the article will be part of a Special Issue entitled “Synthesis and Application of Conjugated Polymers”, it should be pointed out that why D-A polymers are so important in the development of new materials/devices. Therefore the authors should add a short paragraph at the beginning of the Introduction highlighting the impact of these polymers not only in the field of organic photovoltaics. Indeed these polymers found elegant application in other technological fileds as the development of FETs (doi: 10.1038/nmat4785). Most importantly, D-A conjugated polymers became extensively studied for biomedical applications, especially for photothermal therapy (PTT) and its combination with other therapies (doi: 10.1021/acs.biomac.8b01138).

Secondly, the author should highlight in the Introduction the importance of mutual cooperation between organic chemists and material scientists to develop organic solar cells with high performance. The development of the SMOC with the highest efficiency value reported in literature for a single material device (PCE: 5.58%) is one of the most explicate example, because the device based on a tailor-designed D-A polythiophene derivative reached these high-performance thanks to the fine nanostructuration given by the use of electrospinning to obtain ultrathin fibers with aligned polymer chain (doi: 10.1021/acs.macromol.7b00857).

I would recommend adding these small parts in the Introduction section, citing the suggested new and impactful papers;     

Going in details on the specific issues, here some comments are reported:

- The unit of measurements as well as the space within them and the reported values should be standardize throughout the entire manuscript (e.g lines: 81, 82 102, 114, 137, 144, 145, etc etc).

- The unit of measurement reported in all the graphs (TGA, UV-Vis, CV, etc etc) should be reported between square brackets.

- I would suggest including the GPC traces as well as the NMR spectra for P1, P2, P3 and P4, at least in the Supporting Information materials.

- The information regarding the type of the used GPC column should be added.

- Unfortunately, there are no technical information regarding the AFM measurements.  Operation mode (e.g. contact or tapping), cantilever data (material, spring constant, tip curvature), scanning rate, etc etc. All of these pieces of information should be included.

- The authors should add the STD values obtained for the surface roughness (RMS) measurements. Moreover, it should be specified the dimension of the area analyzed to obtain the RMS values.

- The Authors should define every peak/shoulder present in the reported UV-Vis spectra. Moreover, I would be curious to know more about the well-visible peak centered at around 400 nm for “P3” sample in solution.

- Figure 3. Since the CV curves have been shifted to make them visible, the Y axis should be reported in Arbitrary Units. Alternately, the Authors should highlight it in the figure chapter.

- Figure 5. This figure could be quite confusing because some topographies has been collected over a 2 x 2 micrometers while Figure 2A has been obtained scanning a different area different area (3 x 3 micrometers). Please change Figure A showing only a 2 x 2 micrometer area. Moreover, all the axis (X, Y and Z) as well as the figure label (A, B, C, D) should be well visible and with the same font type and font size.

Conclusion

The topic of this manuscript falls within the scope of the Special Issue “Synthesis and Application of Conjugated Polymers” published by Polymers. I like the material and device development/characterization proposed in this paper, moreover the manuscript includes photovoltaic test to support the author claim, anyway I think the manuscript needs a few improvements. I believe the article is of sufficient quality and novelty to meet the Polymers publication standards after a major revision.

Author Response

I would recommend adding these small parts in the Introduction section, citing the suggested new and impactful papers;      

 Thanks for the reviewer’s kind suggestion. According to the suggestion, we have added comments in the introduction section in line 30-33, 39-42 and marked in red. In addition, we have cited suggested paper in the reference [reference 6, 14, 15].   

Going in details on the specific issues, here some comments are reported:

- The unit of measurements as well as the space within them and the reported values should be standardize throughout the entire manuscript (e.g lines: 81, 82 102, 114, 137, 144, 145, etc etc).

  Thanks for the reviewer’s kind suggestion. According to the suggestion, the unit of measurements as well as the space within them and the reported values have been revised.

 - The unit of measurement reported in all the graphs (TGA, UV-Vis, CV, etc etc) should be reported between square brackets.

  Thanks for the reviewer’s kind suggestion. According to the suggestion, we modified the all the graphs (unit of measurement reported between square brackets).

- I would suggest including the GPC traces as well as the NMR spectra for P1, P2, P3 and P4, at least in the Supporting Information materials.

   Thanks for the reviewer’s kind suggestion. According to the suggestion, we have provided the GPC traces and NMR spectra for polymers in the supporting information (Figure S1-S4).

- The information regarding the type of the used GPC column should be added.

  Thanks for the reviewer’s kind suggestion. According to the suggestion, we have added the GPC column in line 107-108 and marked in red.

- Unfortunately, there are no technical information regarding the AFM measurements.  Operation mode (e.g. contact or tapping), cantilever data (material, spring constant, tip curvature), scanning rate, etc etc. All of these pieces of information should be included.

 Thanks for the reviewer’s kind suggestion. According to the suggestion, we have added suggested AFM measurement in line 116-118 and marked in red.

- The authors should add the STD values obtained for the surface roughness (RMS) measurements. Moreover, it should be specified the dimension of the area analyzed to obtain the RMS values.

Thanks for the reviewer’s kind suggestion. According to the suggestion, we have added 3D images of blend films and the corresponding values in supporting information (Figure S5-S8).

- The Authors should define every peak/shoulder present in the reported UV-Vis spectra. Moreover, I would be curious to know more about the well-visible peak centered at around 400 nm for “P3” sample in solution.

 Thanks for the reviewer’s kind suggestion. According to the suggestion, we have defined the peak and commented the well-visible peak centered at around 400 nm for “P3”. The corresponding modification have been marked in red in line 174, 180-182.

- Figure 3. Since the CV curves have been shifted to make them visible, the Y axis should be reported in Arbitrary Units. Alternately, the Authors should highlight it in the figure chapter.

 Thanks for the reviewer’s kind suggestion. According to the suggestion, the Y axis have been reported in Arbitrary Units in Figure 3.

- Figure 5. This figure could be quite confusing because some topographies has been collected over 2 x 2 micrometers while Figure 2A has been obtained scanning a different area different area (3 x 3 micrometers). Please change Figure A showing only a 2 x 2 micrometer area. Moreover, all the axis (X, Y and Z) as well as the figure label (A, B, C, D) should be well visible and with the same font type and font size.

Thanks for the reviewer’s kind suggestion. According to the suggestion, we have provided the modified AFM images in Figure 5.

Round  2

Reviewer 1 Report

My queries have been mostly answered but a few points need further clarification.

(1) Line 149. To justify the comment that higher Mn values were obtained for the present 2T-based polymers than previously reported 3T-based polymers, at least rough (e.g., order of magnitude), but specific, comparison of Mn values is needed. The Mn values range from 30 to 40 kg/mol for the present 2T-based polymers. What were the corresponding values for the 3T-based polymers?

(2) Line 164. The sentence is confusing, which describes that the concentration of polymer was 10^-6 mol/L while the concentration of monomer unit was 10^-5 mol/L. This suggests that degree of polymerization is only 10, which is inconsistent with the Mn values reported.

Author Response

(1) Line 149. To justify the comment that higher Mn values were obtained for the present 2T-based polymers than previously reported 3T-based polymers, at least rough (e.g., order of magnitude), but specific, comparison of Mn values is needed. The Mn values range from 30 to 40 kg/mol for the present 2T-based polymers. What were the corresponding values for the 3T-based polymers?

 Thanks for the kind suggestion. According to the suggestion, we have added the Mn value of our previously reported 3T-based polymers and comparison in Line 154-157, marked in red. 

(2) Line 164. The sentence is confusing, which describes that the concentration of polymer was 10^-6 mol/L while the concentration of monomer unit was 10^-5 mol/L. This suggests that degree of polymerization is only 10, which is inconsistent with the Mn values reported.

Thanks for the kind suggestion. We have re-defined the concentration of monomer unit according to the degree of polymerization of each polymer in line 172-173, marked in red.

Reviewer 2 Report

I appreciate the author effort to respond all my comments and to improve the article. The revised manuscript appears greatly improved and all my concerns have been addressed. Moreover, I appreciate the authors added a few new characterization details and explanations to prove the potential application of the materials. 

The revised version of the article is certainly useful to the scientific community. This is an interesting paper which falls within the scope of Polymers, therefore I recommend this paper for publication without further review.

Author Response

Thanks